# Review: Research Progress of Dairy Sheep Milk Genes

**Ruonan Li** [1,2] , **Yuehui Ma** [1,2] **and Lin Jiang** [1,2,*]

1    National Germplasm Center of Domestic Animal Resources, Institute of Animal Sciences, Chinese Academy of Agricultural Sciences (CAAS), No. 2 Yuanmingyuan West Road, Beijing 100097, China; 82101202343@caas.cn (R.L.); mayuehui@caas.cn (Y.M.)
2    Key Laboratory of Animal (Poultry) Genetics Breeding and Reproduction, Ministry of Agriculture and Rural Affairs, Chinese Academy of Agricultural Sciences (CAAS), Beijing 100097, China
*    Correspondence: jianglin@caas.cn

**Abstract:** The dairy sheep industry is an important but lacking part of the small ruminant industry. For a sheep breeding program, in addition to wool and meat use, sheep milk can also be processed into high-end dairy products such as cheese and milk powder and bring high economic interests for businesses home and abroad. With increasing interest in sheep milk, the content of which is becoming increasingly clearer, people have found that the nutritional value of sheep milk is higher than that of goat milk and cow milk, with abundant fat yield, protein percentage, and mineral contents, which provide a good opportunity for the development of the sheep milk industry. This review will introduce some dairy sheep breeds with the highest milk production worldwide and compare sheep milk nutrition contents with other ruminants' milk. Moreover, genes influencing lactation or mammary gland growth like CSN2, SLC2A2, SCD, and SOCS2, which have been revealed in recent studies to significantly affect milk production and milk composition traits will be discussed. For the SLC2A2 gene, working as an important solute carrier to transport small molecular nutrition from blood to milk and SOCS2 gene mutation as an indicator of mastitis, in addition, other genes have been detected that correlate with milk traits, which will be introduced in the review. Some personal opinions into future sheep milk development will be given in the final part of the text. Although the research of sheep milk genetic factors has achieved some progress in recent years, there is still a long way to go.

**Keywords:** dairy sheep; sheep milk; genes; milk traits; mastitis

## 1. Introduction

Sheep were domesticated in Southwest Asia approximately 11,000 years ago and are now raised around the world for meat, wool, and milk use. There are about 2200 million sheep and goat resources across the world, and a total of 106 sheep breeds, for which 20.8% are intended for dairy production [1]. There are nine main dairy sheep breeds, mostly distributed in Mediterranean and European countries, which include two-thirds of the global dairy sheep population [1]. From domestication to the current pattern of differentiation, domestic species have been influenced by lots of factors such as migration, isolation, and introgression; these have resulted in a very large number of sheep breeds across the world [2]. A large quantity of dairy sheep are distributed in Mediterranean and Black Sea regions, so the sheep milk industry is mainly concentrated in parts of Europe, West Asia, and African countries such as France, Greece, Italy, Spain, Israel, Turkey, and so on; some dairy sheep breeds like the East Friesian sheep, Sarda sheep, Lacaune sheep, Latxa sheep, and Assaf sheep are all high producing sheep breeds known to the world [3]. Previous reports showed that sheep milk has higher fat content, protein content, lactose content, ash content, and total non-fat solids, which have higher nutritional value and can be processed into various delicious milk products and by-products. Moreover, the price of

sheep milk products is two to three times that of cow milk products, and these advantages will bring high economic value for enterprises [4].

For all dairy species, milk yield and compositions are mainly determined by genetic factors, nutrients, and living conditions [5,6]. The genetic factors influence milk yield or milk composition to different extents. In this article, some genes affecting sheep milk performance will be discussed for their significant functions. Considering the growing interest in sheep milk research for improving sheep milk production traits, the aim of this review is to summarize the available literature concerning sheep milk production and find some useful and interesting genes, with particular attention to recent findings. Moreover, the perspective for improving sheep milk production will be given in order to promote the future development of the dairy sheep industry. In the coming years, genetic factors influencing dairy sheep production traits by using genomic and transcriptomics methods to explain milking mechanisms will be highly necessary. Learning more about dairy sheep's' lactation genetic basis could help researchers find some methods to solve the problem of low sheep milk production and thus help sheep milk be spread and consumed by people more widely.

## 2. Descriptions of Some Famous Dairy Sheep Breeds

### 2.1. East Friesian Sheep

Dairy sheep are mostly distributed in dry and warm regions with abundant herbage., and different environments have fostered different breeds of dairy sheep. East Friesian sheep are native to northeast Germany and are currently the world's best milk sheep species [7]. The province of Friesian is the birthplace of both Holstein cows and East Friesian sheep. The two breeds both have the highest milk production among ruminants in the world. Across the world, New Zealand and Australia have the largest number of East Friesian milk sheep and the best milk production performance [8]. Its lactation period lasts for approximately 230 days with milk production of 500–700 kg in a single lactation period [9]. East Friesian sheep is a high-yielding dairy sheep breed generated through long-term good feeding management and genetic improvement and has been imported to many countries. Due to its good milk and meat production performance, the male parent was always imported as to improve local breeds in various countries. Until today, these countries have also developed corresponding dairy sheep breeds that are suitable for local conditions [10].

### 2.2. Lacaune Sheep

The Lacaune breed originates from the Roquefort area of Southern France. It is a main dairy sheep breed in French and has been selected for milk use during the last 40 years and now is a very important dairy sheep breed all over the world [11]. Lacaune ewes produce milk with higher total solids but in slightly less volume than the East Friesians. The milk produced from Lacaune was processed into the famous Roquefort cheese [12]. In the US, the East Friesian and Lacaune sheep are often crossed to produce new dairy breeds that produce milk of higher quality. During the manual milking period, average milk production of these animals was only about 70 L per ewe per lactation. In the 1990s, the milk production had quadrupled to 280 L per year during the Lacaune breeding scheme [13,14]. Today, by the effort of a large-scale strict selection program organized by a French government agency [15], the Lacaune sheep has become one of the world's highest milk producing sheep breeds with milk production of 400–500 kg in a single lactation.

### 2.3. Sarda Sheep

The Sarda sheep is a domestic sheep from Italy and is also a famous milk sheep breed across the world. It is also known by some other names like Cagliari and Sardinian sheep. The breed is indigenous to the island of Sardinia, and raised throughout Italy and other Mediterranean countries, particularly in Tunisia. Currently, the Sarda sheep is considered to be the one of the best Italian sheep breeds for the production of sheep milk. In Italy,

most milk of ruminant animals is used to make pecorino sarda cheese, which is extremely popular in European countries, and Sardinia is one of the main ovine milk providers in southern Europe, with more than three million Sarda sheep distributed there [16]. The average single test-day milk yield of Sarda sheep is 1.47 kg/d, and the average fat and protein content is 6.35 and 5.32%, respectively [16]. For the total milk yield, it reaches 376 kg in an average lactation [17].

### 2.4. Latxa Sheep

The Latxa sheep is a domestic sheep from Spain. It is a dairy sheep breed raised mainly for milk production in its native area. These sheep are mostly distributed in the Basque Country and Navarre. They are known for high quality milk production. The unpasteurized milk is often used for producing Roncal cheeses. Since 1982, a breeding scheme for Latxa sheep based on milk recording has been run in a large group of farms [18], these resulted in the highly improved genetic basis in promoting milk yield of Latxa sheep. In the past 40 years, Latxa sheep have been traditionally selected with the introduction of modern quantitative genetics and breeding methods, a genetic improvement program for this breed was established in 1981 [19,20]. Initially, the goal was to increase milk production in order to improve the profitability of the flocks. Later, milk compositions and milk quality were also included into the selection criteria of the program, together with resistance to some diseases. This promoted Latxa sheep and led to improvements in certain traits [21], and now the daily milk yield of Latxa sheep reaches 2.69 kg/d.

### 2.5. Awassi Sheep

Awassi dairy sheep are now mainly distributed in the Middle East regions, which can be used for milk, meat, and wool production [22]. Awassi sheep were obtained through systematic cultivation in Israel, whose lactation period reaches 214 days, and the milk yield of each lactation period is approximately 500 kg nowadays [23]. Milk production keeps increasing until reaches a peak at the third week of lactation, then it begins to decline. The outstanding characteristic of this breed is the tolerance of extreme temperatures and terrible feeding conditions [24]. Through centuries of natural and selective breeding, nowadays it has become the highest milk producing breed in the Middle East. Genetic improvement and selection programs for Awassi sheep have resulted in the development of improved Awassi breeds, such as Afec Awassi by introducing the FecB Booroola gene into Awassi breeds. Now, the improved Awassi sheep are characterized by producing the highest volume of milk, with the highest fertility and twinning rate compared with previous populations [25].

### 2.6. Assaf Sheep

Assaf dairy sheep is a domestic sheep breed cultivated in Israel. Researchers of the Israeli Agricultural Research Organization started the improvement project in 1955, aiming to improve the fecundity of the Assaf sheep breed. The Assaf sheep is a combination of 3/8 East Friesian and 5/8 Awassi blood [26]. It is raised as a dual-purpose animal for both meat and milk, but they are mainly raised for milk production. Assaf dairy sheep produce 300–400 kg of milk during a lactation period of 170–200 days [27]. In Spain, it is mainly raised in the Castila and Leon regions, where the population now reaches 900,000, and 95% of Assaf sheep milk is used to process high-quality cheese, of which 70% of cheese production is exported to other countries. It is an important dairy sheep breeds in Spain and Mediterranean regions, owing to its good production traits, and has been imported to many other countries [28].

In Table 1, we concluded some milk traits of this dairy sheep breeds, as seen below.

**Table 1.** Some milk performance of East Friesian, Lacanue, Sarda, Latxa, Awassi, and Assaf sheep.

| Breed | Total Milk Yield/Lactation (kg) | Lactation (Day) | Average Milk Yield per Day (kg) | Average Protein Percent (%) | Average Fat Percent (%) | Reference |
|---|---|---|---|---|---|---|
| East Friesian | 700 | 300 | 2.29 | 5.35 | 6.50 | Thomas, D.L. et al., 2014 |
| Lacanue | 454 | 200 | 2.50 | 5.00 | 5.93 | Alba, D.F. et al., 2019 |
| Sarda | 376 | 150 | 2.50 | 5.32 | 6.35 | Bittante, G. et al., 2017 |
| Latxa | 446 | 147 | 3.03 | 5.75 | 6.09 | Juste, R.A. et al., 2020 |
| Awassi | 460 | 214 | 3.71 | 5.15 | 6.86 | Nudda, A. et al., 2020 |
| Assaf | 506 | 173 | 2.90 | 6.00 | 6.50 | Pollott, G.E. et al., 2004 |

From the table, we can see the East Friesian sheep have the highest milk production and Sarda sheep have the lowest milk production per lactation. East Friesian sheep have the longest lactation days and Latxa sheep have the shortest lactation days. Moreover, Awassi sheep have the most milk yield per day and largest fat percent volume and Assaf sheep have the highest average protein percent.

## 3. Comparison of Cow, Goat, and Sheep Milk Contents

Compared with other kinds of milk, the content of fat, protein, lactose percent, and vitamin and mineral contents of sheep milk are higher than those of goat milk and cow milk [29]. Park [30] found that the total fat content of sheep's milk was 51% and 54% higher than that of goat's milk and cow milk, respectively, and sheep's milk was richer in unsaturated fatty acids. Unsaturated fatty acids in food are the source of high-quality animal fat that is necessary to the human body and plays a very important role in regulating blood lipids, clearing thrombosis, enhancing immunity, improving eyesight, and strengthening the brain [31]. Moreover, the trans fatty acid content in sheep milk was 6 times of goat milk, and the unsaturated fatty acid content of sheep milk was also three times of other small ruminants in Schroeder. In previous research, a study was performed to detect the fatty acid content in sheep and goat milk, and an amazing result was discovered: the 4, 7, 13, 16, 19-C22:6 fatty acid was not detected in goat milk, while in sheep milk the content was 18.31% [32]. In addition to fat percent, the protein content of sheep milk was also approximately two times that of goat milk, especially for the casein content, which is beneficial to human health [29]. Moreover, previous studies have showed that the mineral content of calcium, magnesium, iron, copper, zinc, and manganese in sheep colostrum was much richer. In addition, the content of vitamin C and vitamin B in sheep milk was three times that of goat milk and two times that of cow milk [33]. The detailed milk contents of cow, goat, and sheep milk are shown in Table 2.

**Table 2.** The comparison of cow, goat, and sheep milk compositions.

| Content | Unit | Cow Milk | Goat Milk | Sheep Milk | Reference [2] |
|---|---|---|---|---|---|
| Fat | g/kg | 36 | 38 | 79 * | Pietrzak-Fiećko, R. et al., 2020 |
| Protein | g/kg | 33 | 35 | 62 * | Roy, D. et al., 2020 |
| Lactose | g/kg | 46 | 41 | 49 | Roy, D. et al., 2020 |
| Ash | g/kg | 7 | 8 | 9 | Pietrzak-Fiećko, R. et al., 2020 |
| Non-fat solid | g/kg | 79 | 76 | 111 * | Pulina, G. et al., 2018 |
| Iodine | mg/100 g | 0.5 | 0.41 | 0.47 | Slovenian, P. et al., 2014 |
| Phosphorus | mg/100 g | 96 | 87 | 160 * | Park, Y.W. et al., 2000 |
| Cuprum | mg/100 g | n/a [1] | 0.013 | 0.019 | Stocco, G. et al., 2018 |
| Calcium | mg/dL | 120 | 130 | 180 * | Pulina, G. et al., 2018 |
| Zinc | mg/dL | 0.62 | 0.69 | 0.58 | Claumarchirant, L. et al., 2015 |
| Sodium | mg/dL | 49.3 | 35.9 | 52.1 | Singh, M. et al., 2019 |
| Magnesium | mg/dL | 12 | 16 | 21 | Khan, I.T. et al., 2019 |
| Potassium | mg/dL | 148 | 184 | 179 | Khan, I.T. et al., 2019 |
| VitaminA | mg/100 g | 0.04 | 0.05 | 0.08 | Burrow, K. et al., 2019 |
| Vitamin B$_6$ | mg/100 g | 1.09 | 1.01 | 1.75 * | Ochoa-Flores, A.A. et al., 2021 |
| Vitamin C | mg/100 g | 0.09 | 1.29 | 4.16 | Caboni, P. et al., 2019 |

[1] Data not available; * Means nutritional values of sheep milk are significantly higher than cow milk and goat milk; [2] Data come from these references.

## 4. Genes Affecting Sheep Milk Performance

Sheep milk traits have many different phenotypic diversities that have attracted the attention of various researchers worldwide to investigate its hidden genetic and genomic determinism. More and more genomic-based investigation approaches have been used in genetic comparisons of different populations [34]. This can identify genes regulating the investigated traits, and with the advancement of agricultural technologies, the understanding of genetic bases underlying target traits has increased drastically [35,36]. Many studies with diverse breeds of dairy sheep have indicated casein gene (CSN) as a promising candidate gene for milk compositions traits [37,38]. Moreover, the SCD gene, SOCS2 and SLC2A2 gene were also chosen as significant candidate genes affecting milk fat, milk protein, or milk quality in recent studies [39–41]. These genes will be introduced and discussed in detail below, and some of the latest research outcomes towards finding and understanding high-potential genes of sheep milk are listed in Figure 1. These potential genes will also be introduced in the main text.

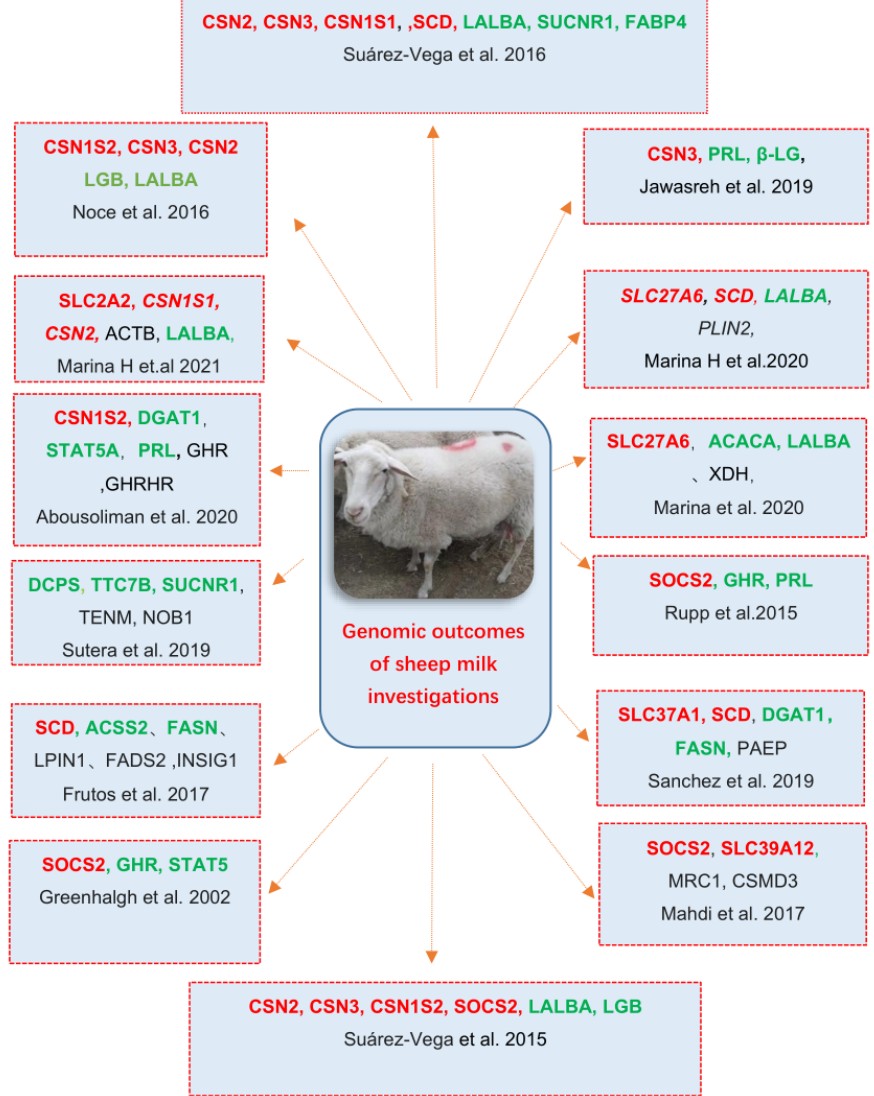

**Figure 1.** Genomic outcomes of sheep milk, including some potential genes generated from previous investigations of different researches. Repeatedly described genes are marked in red and green colors. Red indicates the significant genes SLC2A CSN2, SCD, and SOCS2, as discussed clearly in the text. Green indicates the genes detected that are important to milk traits in previous studies, but with little research.

## 5. SLC2A2 Gene—Transport Nutrition from Blood to Milk

SLC2A2 (Solute Carrier Family member 2) is a member of solute carrier (SLC) family. The solute carrier family is made up of different transporter families that include ion channels and exchangers, its passive transporters SLC2A2 is located on chromosomes 1 and belongs to the passive transporters, which are related to the biological process of carbohydrate metabolic process and insulin secretion regulation [42]. Many researchers have found this gene is related to milk synthesis and milk lactation [43–45]. In H. Marina's research [46], GWAS and post -GWAS were done in two dairy sheep breeds of 2020 ewes, and different analysis was performed to detect candidate genes associated with milk production and composition traits, as well as cheese making traits. Among all detected SNPs and QTLs, 11 QTL previously described by [47] on OAR24 (OAR24:26228200–38615161 bp) related to milk fat content traits were identified here for both Assaf (in relation to FP) and Churra breeds. A total of 60 genes were prioritized based on cheese-making traits and milk production traits. Among these genes, SLC2A2 was found to be a significant candidate gene. Through GO and KEGG analysis, this gene was found to participate in carbohydrate metabolic processes and their transmembrane transport, in mammary gland, as the SLC2A2 protein works as a transport carrier to transport glucose and other soluble small molecules [42]. These soluble small molecules then become nutritional components in ovine or bovine milk in the mammary gland.

Besides, in secretory tissues such as the mammary gland, $Zn^{2+}$ transporters of SLC families are involved in specific functions like insulin synthesis and the secretion of some digestive proenzymes, transferring nutrients from blood to milk. Defective $Zn^{2+}$ metabolism in these tissues will cause diabetes and cancer, and stop $Zn^{2+}$ secreting into milk [48]. Other solute carrier genes, such as the SLC27A6 and SLC37A1 genes, have also been revealed to be associated with milk traits such as the fatty acid composition and the mineral content of bovine milk in previous reports [49,50]. Therefore, the function of these SLC family genes still have a lot to be further investigated.

## 6. CSN2 Gene—Benefit for the Lactose Intolerance

Casein (CSN) is the main protein component of milk in most milking species and accounts for about 80% of total milk protein [51]. Its functions include transporting calcium and phosphate to milk to provide enough calcium and phosphorus for bone formation in young lambs and to meet amino acid requirements. Casein includes α S1-casein (CSN1S1), α S2-casein (CSN1S2), β-casein (CSN2), and k-casein (CSN3) [37,52,53]. Sheep casein contains 45% β-casein, represented by two phosphorylated forms, β1-casein and β2-casein, which have similar amino acid compositions as bovine β-caseins and have a significant effect on milk protein [54].

Interestingly, people also found that milk containing A2 β-casein (CSN2) only causes fewer and slighter symptoms in lactose intolerance (LI) than milk containing both A1 and A2 β-caseins [55,56]. Researchers conducted a single-meal study to evaluate the effect of milk containing A2 β-casein proteins on the gastrointestinal (GI) tolerance. Subjects drunk each of four types of milk (milk containing A2 β-casein protein only, conventional milk, Jersey milk, and lactose-free milk) after overnight fasting. In an analysis of 25 LI subjects, the total symptom score for abdominal pain was lower for drinking milk containing A2 β-casein only compared with the other three kinds of milk [57]. They further investigated the reasons for the symptoms, and the results discovered that those with LI drinking milk containing A2 β-casein only produced lower content of hydrogen and meanwhile increased the time for milk to leave the colon, which promoted the digestion of milk, and then, abdominal pain, bloating, and flatulence were alleviated or even disappeared [58]. This finding is very useful for dairy sheep research, which could be used to produce A2 β-casein sheep milk, and lactose intolerance will be not a concern regarding the occurrence of abdominal pain and lactose intolerance.

## 7. SCD Gene—Promote Unsaturated Fatty Acids Yield

For many years, people have paid important attention to unsaturated fatty acids in preventing the risk of cancer. Fatty acids are one of the most important bioactive components in mammalian milk [59]. Due to their high nutritional value and function on the physicochemical processes of the body, they are necessary for the development of the nervous system and the growth of a young body. These discoveries increased the demand of sheep milk, as sheep milk fat contains several components that provide humans great health benefits, notably a large quantity of unsaturated fatty acids and conjugated linoleic acid (CLA) [60]. Most unsaturated fatty acids in ruminant milk are synthesized in the mammary gland by the action of an enzyme called stearoyl-CoA desaturase (SCD) working on circulating vaccenic acid synthesis. Five years ago, Suárez-Vega [61] found SCD gene as a highly expressed gene by doing RNA-seq in two Spanish sheep breeds, four Churra sheep, and four Assaf Sheep. He found the FPKM values of the SCD gene were higher in Churra than in Assaf sheep, and that Churra sheep have higher milk fat content. These two results are consistent, because the SCD gene has higher FPKM in Churra and it participates in the synthesis of body fat. Higher SCD gene expression quantity promotes the synthesis of milk fatty acids and contributes to higher milk fat in Churra sheep [61,62]. In addition, Gu M [39] combined mRNA profile analysis, quantitative PCR analysis, and electromobility shift assay to explain the role of a SNP in SCD gene transcription and its effects on milk fat traits. The SNP g.133A>C [63] was carried out on a group of 303 buffaloes, and the association study with milk fat traits uncovered a favorable effect of allele C. Heterozygous genotype (AC) had the highest contents of monounsaturated fatty acids, branched-chain fatty acids, oleic acid (C18:1 cis-9), and odd-chain acids, and the lowest percent of saturated fatty acids, thrombosis, and atherosclerosis. These data demonstrated the role of SNP g.133A>C in the SCD promoter and its association with fat acids synthesis, which provides foresight insights into the genetic background of lipid metabolism and protein metabolism, including the possibility of future selection of alleles with quantitative or qualitative favorable effects.

## 8. SOCS2—An Indicator of Mastitis in Mammary Gland

Mastitis is an infectious disease mainly caused by staphylococcus aureus and streptococcus invading the mammary gland. Genetic control of susceptibility to mastitis has been widely proved in dairy ruminants, but the genetic basis underlying this disease is still largely unknown. Initially, Rachel Rupp et al. [64] found the identification of a mutation in the Suppressor of Cytokine Signal 2 gene (SOCS2) by implementing a genome-wide association study. This mutation was shown to cause a deficiency on functional activity of the SOCS2 protein, which suggested an impairment of negative control on the JAK/STAT signal pathways in mastitis animals [40,65]. QTLs associated with SCC were mapped by performing a genome scan on 26 autosomes in 1009 dairy sheep, one highly significant QTL on chromosome 3 was similarly mapped by two associations and linkage analysis. After further investigation, only one out of 207 SNPs was located in the coding region of a gene SOCS2 with a non-synonymous change in an amino acid. Replacing of A to T caused the amino acid substitution at position 96 (p.R96C). Analysis of variance confirmed that the T mutation -constituted genotype showed a dramatic increase in somatic cell counts and milk yield, but also significantly decreased milk fat content. The mutation was found to located within the SH2 domain. This change can disturb the tri-dimensional structure of the domain and destroy the protein functions [66]. Many studies have reported the essential role of SOCS2 proteins in the regulation of cytokine, hormones such as prolactin, growth hormone, and erythropoietin, as a negative control of the cytokine involved signal pathways [65–67]. When sheep were infected with mastitis, high milk SCC showed chronically rising white blood cells in milk of animals carrying the p.R96C mutation in SOCS2 gene. This mutation also causes positive expression of prolactin, with milk production increased consequently. Similarly, Harris et al. [68] showed that SOCS2 is a key regulator of the prolactin signal pathway and balanced the shortage in mammary gland development

produced in two prolactin deficient models, thereby demonstrating the role of SOCS2 in the control of mammary gland growth, development, and subsequent milk production [69].

## 9. Other Detected Genes Related to Milk Traits

Since 2017, various genomic-based studies have been conducted to figure out the genetic basis on sheep milk to find the specific genomic variant(s) contributing to sheep milk production [50,70,71]. For example, Sutera, A.M. carried out GWAS for milk production traits in 469 Valle del Belice sheep using repeated measures, and the research found SNP rs425417915, which is located in an intronic region of TTC7B (tetratricopeptide repeat domain 7B) gene, to be associated with both fat percent and protein percent, playing a critical role in lipid metabolism in cattle and was reported to be associated with fat yield (FY) composition in sheep [72]. Additionally, SNP rs417079368 associated with fat percent and protein percent in the analysis, was higher than genome-wide significant threshold and has been identified at 0.37 Mb of the SUCNR1 (succinate receptor 1) gene. Research showed SUCNR1 is related to the pathways concerned with cheese traits in sheep milk and it is expressed in lots of tissues and organs, particularly in the mammary gland [61].The third SNP, SNP rs398340969, was associated with MY and PY, and it is mapped within an intronic region of the DCPS gene, which is found involved in the cellular response to vitamin K3 in humans, and highly expressed in mammary tissue [72]. Three SNPs were associated with more than one trait. It is well known that genetic correlation exists among different milk production traits, and it would be interesting to investigate the allelic substitution effects of these SNPs on milk production traits considered in the study.

Similarly, three statistical methods, GWAS, ridge-regression BLUP, and Bayes C, were used to identify SNPs related to three milk production traits (milk yield, fat yield, and protein yield) in a crossbred dairy sheep population. The results suggested that about 100 significant SNPs contributing to the three milk production traits [73]. Similarly, one SNP was located on the TTC7B gene and the GWAS results found it was related with fat yield. While other relevant candidate genes affecting milk production includes STAT5A, DGAT1, FABP3, ACACA, FSN, ACSS1, and ACSS2 have also been reported previously in other articles [74–81]. In all the detected genes above, however, we found little or even none of the related research or explanations of how these genes work on milk quantity or quality traits, which thus need to be further investigated.

## 10. Personal Insight into Future Sheep Milk Research

Currently, Chinese sheep are mainly used for meat and wool production, and sheep milk is extremely important to the sheep industry due to its high nutritional value. However, the dairy sheep industry in many countries remain blank. Therefore, the future development of sheep enterprises with milk production will be a new direction for the development of animal husbandry. Although some genes have been proven to be associated with milk performance, the detailed regulating pathways and gene network of candidate genes in sheep still need to be explored. In the past, molecular research techniques, such as quantitative trait locus (QTL) mapping, molecular assistant selection (MAS), and linkage disequilibrium analysis (LD), were used to discover many QTLs, variations, and phenotype associations. However, these methods are complex and have low accuracy. Thus, new genomic prediction methods, such as Whole Genome Resequencing analysis, High-Density SNP chips, RNA-SEQ, and Chip-SEQ [82–85] can be used to select genes related to complex traits, which will be more accurate and faster. A Genome Sequence Scan Analysis or a Genome-Wide Association Study to analyze the dairy sheep's critical genes and mutations associated with milk performance will be more efficient and convincing for their high-density mapping rate and the development of dairy sheep genetic improvement will proceed faster.

Furthermore, for genome editing as a multi-function method that can be used to intervene the expression of detected genes, and in various animal species, CRISPR–Cas9 has been used to verify important traits that are economically, biomedically, and agriculturally

significant [86]. These aim to investigate their biological roles in association with milk phenotypes and other economic traits (increased milk yield), especially if genes, such as SCD and SLC2A2, are positively associated with milk performance. After gene editing, other omics studies can be performed to track the changes in these processes [87]. Investigating the gene knockout, gene interference, or gene overexpression effect on the traits of milk accumulation will also be of great importance in subsequent confirmatory experiments [88]. By conducting gene editing technology, the phenotypic characteristics of gene-edited sheep can be detected when compared to their wild-type counterparts, therefore, these methods could provide novel insights into the effect of these variants and reveal the causative mutation(s) of the targeted gene working on special traits.

## 11. Conclusions

The global sheep milk market has maintained a rapid growth trend in recent years, but the total stock and research on dairy sheep in some developing countries are still lacking. The development of the dairy sheep industry will promote the development of the upstream and downstream of dairy sheep industry, such as fodder, herbage, milk powder, and meat products, and milk processing and selling will create regional characteristic industries, which will provide more profit for companies and individuals. So, once domestic enterprises pay more attention to sheep milk genetic research and dairy sheep breeding programs, sheep milk is expected to be extremely popular across the world. In our article, we have introduced the specific function of four genes, SLC2A2, CSN2, SCD, and SOCS2, and among these genes, the SLC2A2 gene takes charge of transporting soluble molecules from blood to mammary gland, the CSN gene plays a role in casein synthesis, the A2-CSN2 gene shows resistance to lactose intolerance, the SCD gene participates in the production of unsaturated fatty acid and fat percent, and the SOCS2 gene mutation works at an indicator of high SCC and mastitis in dairy sheep, which is also proved to be a negative regulator factor participates in the JAK/STAT pathways. In the future, there are more genes waiting for us to find and figure out their function in milk generation. For most people concerned, another important and meaningful step is to find some genes related to the flavor of sheep milk, as some people do not like drinking sheep milk owing to the strange smell of it. If we can find some genes regulating the traits of the strange flavor, we can help industries develop sheep without such smells and tastes.

**Author Contributions:** R.L., conceptualization and writing—original draft; L.J., methodology, correction, and supervision, Y.M., resources, supervision, and project administration. All authors have read and agreed to the published version of the manuscript.

**Funding:** This research received no external funding.

**Institutional Review Board Statement:** Not applicable.

**Informed Consent Statement:** Not applicable.

**Data Availability Statement:** The data that support the findings of this study are available from the corresponding author upon reasonable request.

**Conflicts of Interest:** The author declares that the review was written without any commercial or financial conflict of interest.

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
