# Peer review of "Review: Research Progress of Dairy Sheep Milk Genes"

_agriculture, doi:10.3390/agriculture12020169_

Round 1

Reviewer 1 Report

The subject of the manuscript is actual and describes an interesting topic. The title tells us about the research progress of dairy sheep milk genes however, the article only gives short explanation of the genes, but not of the history of discovering them.

On the other hand, historical description of the breeds was a little bit too long.

Lines 13 to 16: phrase "sheep milk" 4 times in one sentence

Line 21: "a transporter to transport"- please give a synonim

Line 33: 9 main sheep? (breeds)

Line 48: I would rephrase the sentence. Heritability estimations account for 0.2-0.35 for milk yield and composition --> so not mainly. Maybe add that nutrition and optimal environment is a must to express genetic potential.

Line 69: "Produce the highest milk production": please rephrase

Line 85: "Human milking period" - confusing 

Lines 128-129: "sheep genetic improvement  .... improved Awassi breeds"- to where?

Line 145: Table 1: It's a bit confusing that authors made statistics for 6 averaged data? 

Line 148: Most of the data... database size or number of references should be mentioned somehow.

Line 175: Add the applied statistics

Lines 178 to 190: Font type is different than the other paragraphs

Line 282: Rupp et al.

Lines 325 to 333: Please add correlation coeffitiens

Reviewer 2 Report

The manuscript “Review: Research progress of milk sheep milk genes” presents a relevant subject. However, I have some suggestions and considerations:

Line 49 -factors, and a small part reason for the nutrition impact and living conditions, change to: factors, and to a lesser extent the nutritional impact and living conditions

Line 77 - cultivated or developed? different dairy sheep breeds that were suitable for local conditions

Table 1. Some milk performance of East Friesian, Lacanue, Sarda, Latxa, Awassi and Assaf sheep - important changes:

Line 146 - 1There is a significant difference on the total milk yield of six breeds and Line 147- 2There is no significant difference on the average protein percent and fat percent among six breeds.

To state that there is a statistical difference between the values, you must explain the statistical procedure adopted, as well as the deviations from the means (standard error, standard deviation, coefficient of variation). Otherwise, you cannot mention statistical differences.

Line 149 - Most data are cited from these references. As well? Weren't they all?

Correct and improve the presentation of the bibliographic references in Table 1.

Line 152 – Compared with other kinds of milk, the dry matter content and fat, protein and lac, change to: Compared with other types of milk, the content of total solids and fat, protein and

Line 383 – favour, change to flavour

Round 2

Reviewer 1 Report

I can accept the manuscript in its present form.